# Diagnostic Roles of α-Methylacyl-CoA Racemase (AMACR) Immunohistochemistry in Gastric Dysplasia and Adenocarcinoma

**DOI:** 10.3390/medicina60091475

**Published:** 2024-09-09

**Authors:** Jung-Soo Pyo, Kyeung-Whan Min, Ji-Eun Choi, Dong-Wook Kang

**Affiliations:** 1Department of Pathology, Uijeongbu Eulji Medical Center, School of Medicine, Eulji University, Uijeongbu-si 11759, Republic of Korea; kyueng@eulji.ac.kr; 2Department of Pathology, Chungnam National University Sejong Hospital, 20 Bodeum 7-ro, Sejong 30099, Republic of Korea; b612elf@cnuh.co.kr; 3Department of Pathology, School of Medicine, Chungnam National University, 266 Munhwa Street, Daejeon 35015, Republic of Korea

**Keywords:** AMACR, immunohistochemistry, gastric dysplasia, gastric adenocarcinoma

## Abstract

*Background and Objectives*: This study aimed to elucidate the diagnostic role of α-Methylacyl-CoA racemase (AMACR) immunohistochemistry in gastric dysplasia and adenocarcinoma. *Materials and Methods*: Immunohistochemistry for AMACR was performed on 39 gastric dysplasia and 40 gastric adenocarcinoma cases. The expression patterns of AMACR were investigated and divided into luminal and cytoplasmic expression patterns in the gastric lesions. In addition, correlations between AMACR expression and patient age, sex, and tumor size were evaluated. *Results*: AMACR was expressed in 26 of 39 cases of gastric dysplasia (66.7%) and 17 of 40 cases of gastric adenocarcinomas (42.5%). The AMACR expression rates in high- and low-grade dysplasia were 80.0% and 52.6%, respectively. A detailed analysis of the expression patterns revealed that the luminal expression pattern was significantly higher in low-grade dysplasia than in high-grade dysplasia and gastric adenocarcinoma (*p* < 0.001). The cytoplasmic expression pattern, without luminal expression, was predominant in high-grade dysplasia and gastric adenocarcinoma. In addition, the rates of loss of expression in the overall area were 15.1 ± 23.9%, 49.0 ± 29.9%, and 59.0 ± 32.2% in low-grade dysplasia, high-grade dysplasia, and gastric adenocarcinoma, respectively. The negative rate of low-grade dysplasia was significantly lower than that of high-grade dysplasia and gastric adenocarcinoma (*p* < 0.001 and *p* < 0.001, respectively). *Conclusions*: AMACR is a useful diagnostic marker for differentiating low-grade dysplasia from high-grade dysplasia and gastric adenocarcinoma. Luminal or cytoplasmic expression patterns and the extent of loss of expression are important for differentiation.

## 1. Introduction

Gastric dysplasia is a neoplastic lesion with no stromal invasion [1]. Gastric dysplasia is classified as low- or high-grade dysplasia, based on the degree of cellular abnormality [1]. In a previous study, the prevalence of low- and high-grade dysplasia and adenocarcinoma in the stomach was observed in 1–2%, 0.1–0.2%, and 0.4–0.5% of all participants, respectively [2]. Stomach biopsy samples are the most commonly encountered specimens in daily practice. When adenoma or adenocarcinoma is diagnosed, further treatment is required. Diagnosis can be difficult in cases with fewer lesions in the specimen. Moreover, non-neoplastic changes such as inflammation and regeneration exhibit cellular atypia and mitotic figures [1]. Thus, differentiating between non-neoplastic changes and dysplasia is crucial in daily practice. Distinguishing between low- and high-grade dysplasia also presents challenges. Furthermore, achieving agreement between pathologists in diagnosing low- and high-grade dysplasia is difficult [3]. While immunohistochemistry may offer assistance, only a few markers have demonstrated a definitive diagnostic value in differentiating gastric lesions. α-Methylacyl-CoA racemase (AMACR) is a highly specific marker for prostate cancer [4]. AMACR was found to be overexpressed in primary and metastatic prostate cancer and high-grade intraepithelial neoplasms, but not in normal tissues [4]. Described as a cytoplasmic protein, AMACR plays a significant role in the oxidation of branched-chain fatty acids and their derivatives [4]. Jiang et al. investigated the expression of AMACR in various normal tissues and malignant tumors, including the stomach [5]. AMACR is expressed in 25% and 75% of gastric and colorectal adenocarcinomas [5,6]. Interestingly, AMACR was also found to be overexpressed in some normal tissues, including hepatocytes, bronchi, gallbladder epithelial cells, and renal tubules [5]. However, in contrast, other normal human tissues, including the stomach, showed no expression [6]. Colon cancers with good and moderate differentiation are mostly classified as AMACR positive [6]. Researchers have investigated the role of AMACR in the differentiation of gastric lesions [3,7,8]. The expression rate of AMACR varies from 51.5% to 62.9% in gastric adenocarcinoma [7,8]. Similarly, the expression rate of AMACR in gastric dysplasia ranges from 43.5% to 83.3% [3,7,8]. However, these findings suggest that AMACR lacks a significant value in the differential diagnosis of gastric lesions. AMACR expression has also been identified in dysplasia associated with Barrett’s esophagus and inflammatory bowel diseases [9,10]. However, there is a notable difference in AMACR expression between low- and high-grade dysplasia in Barrett’s esophagus [9]. Moreover, AMACR expression is significantly higher in the stomach in high-grade dysplasia than in low-grade dysplasia (76.0% vs. 4.5%) [3]. This indicates that AMACR can potentially aid in the differential diagnosis of gastric lesions. However, identifying meaningful differences between gastric lesions based on existing studies remains challenging [3,7,8]. This study aimed to elucidate the diagnostic role of AMACR immunohistochemistry in gastric dysplasia and adenocarcinoma. In this study, we focused on three main aspects. First, the expression rate of AMACR in various gastric lesions was examined. Second, we compared the AMACR expression patterns among gastric lesions. Third, the presence of AMACR expression in normal mucosa adjacent to the lesions was analyzed.

## 2. Materials and Methods

### 2.1. Patients and Tissue Array Methods

The present study included 79 patients who underwent endoscopic submucosal dissection or gastrectomy at the Eulji University Medical Center from 1 April 2021 to 31 August 2023. The diagnostic criteria for these patients were low-grade dysplasia (*n* = 20), high-grade dysplasia (*n* = 19), and adenocarcinoma (*n* = 40). We reviewed the pathological records and hematoxylin and eosin (H&E) slides to gather relevant data. The information collected included age, sex, tumor size, and diagnosis. The study protocol was reviewed and approved by the Institutional Review Board of the Eulji University Hospital (approval number: UEMC 2023-10-011) on 13 November 2023.

### 2.2. Immunohistochemical Staining

For immunohistochemistry, 4 μm thick sections were cut from each paraffin block, deparaffinized, and dehydrated. Immunohistochemical staining was conducted following the compact polymer method using a VENTANA benchmark ULTRA autostainer (Ventana Medical Systems, Inc., Tucson, AZ, USA). The sections were then incubated with anti-AMACR (clone 13H4; Dako, Carpinteria, CA, USA). Visualization was performed using an OPTIVIEW universal 3,3′-diaminobenzidine kit, according to the manufacturer’s instructions (Ventana Medical Systems, Inc.). To ensure the reaction specificity of the antibody, negative control staining without the primary antibody was also conducted. All immunostained sections were lightly counterstained with Mayer’s hematoxylin.

### 2.3. Evaluation of Immunohistochemistry

AMACR immunoreactivity was detected in the cytoplasm. The intensity of protein expression in immunohistochemically stained samples was scored from 0 to 3 (0 = negative, 1 = weak, 2 = moderate, and 3 = strong). The percentage of positively stained cells was determined using a scoring system that ranged from 0 to 4 (0 = negative; 1 = ≤25%; 2 = 26–50%; 3 = 51–75%; and 4 = 76–100%). The immunoreactive score (IRS) was subsequently calculated by multiplying the staining intensity score by the percentage of positively stained cells [11]. Based on IRS, immunohistochemical staining was classified as negative (IRS 0–4) or positive (IRS > 4). The rate of loss of AMACR expression was assessed by calculating the proportion of regions with no AMACR expression relative to the entire lesion. To assess heterogeneity, AMACR expression in a hotspot (area = 0.785 mm^2^) was evaluated. The hotspot area was selected as the highest level of immunoreactivity on the H&E slide after scanning at a medium power (×100). In addition, AMACR cytoplasmic expression was subdivided to examine the luminal pattern separately. Two independent researchers (J.S. Pyo and D.W. Kang) evaluated the immunohistochemical results with the naked eye under the microscope. In cases of discrepancy, the results were reviewed and the two researchers reached a consensus.

### 2.4. Statistical Analysis

Statistical analyses were conducted using SPSS software version 22.0 (IBM Co., Chicago, IL, USA). The χ^2^ test was used to assess the significance of the correlation between AMACR expression and sex. Correlations between the luminal and cytoplasmic expression patterns of AMACR were also evaluated using Fisher’s exact test. Comparisons between AMACR expression, age, and tumor size were analyzed using a two-tailed Student’s *t*-test. The negative rates of AMACR expression between gastric lesions were also evaluated using a two-tailed Student’s *t*-test and the Kruskal−Wallis test. The results were considered statistically significant at *p* < 0.05.

## 3. Results

### 3.1. AMACR Expression in Gastric Dysplasia and Adenocarcinoma

AMACR immunohistochemical staining was conducted, and the representative images are shown in Figure 1. AMACR expression was observed in 26 of 39 cases of gastric dysplasia (66.7%) and in 17 of 40 cases of gastric adenocarcinomas (42.5%; Table 1). 

In the gastric dysplasia cases, AMACR expressions were evaluated and divided into low- and high-grade dysplasia. AMACR expression was noted in 16 of 20 low-grade dysplasia cases (80.0%) and in 10 of 19 high-grade dysplasia cases (52.6%). In addition, AMACR cytoplasmic expression was subdivided to examine the luminal pattern separately. Upon further analysis, the luminal pattern of AMACR expression was found to be more prevalent in low-grade dysplasia than in high-grade dysplasia or gastric adenocarcinoma (*p* < 0.001; Table 1). The luminal pattern of AMACR expression was identified in 14 of the 16 positive cases with low-grade dysplasia (87.5%). However, the luminal pattern of AMACR expression was observed in 30.0% and 6.3% of high-grade dysplasia and adenocarcinoma cases, respectively. There were significant differences in the AMACR expression patterns between low-grade dysplasia and high-grade dysplasia and between low-grade dysplasia and adenocarcinoma (*p* = 0.003 and *p* < 0.001, respectively). However, there was no significant difference between high-grade dysplasia and adenocarcinoma (*p* = 0.197), according to Fisher’s exact test. Next, the correlations between AMACR expression and clinicopathological parameters such as age, sex, and tumor size were evaluated. In low-grade dysplasia, AMACR expression was significantly correlated with younger patients and smaller tumor size (*p* = 0.041 and 0.025, respectively; Table 2). However, there was no significant correlation between AMACR expression and sex in the low-grade dysplasia group (*p* = 0.530). In addition, there was no significant correlation between AMACR expression and clinicopathological parameters, such as age, sex, and tumor size, in the high-grade dysplasia and adenocarcinoma groups (Table 2). Furthermore, AMACR expression in hotspots was assessed. The positivity rate of AMACR expression in the hotspots was higher than that in the overall area across all lesion types (Table 3). In low-grade dysplasia cases, the AMACR expression rate reached 100% in hotspot areas. However, upon evaluating hotspots, the negative rates of AMACR expression in high-grade dysplasia and adenocarcinoma were found to be 37.7% and 35.0%, respectively.

### 3.2. Loss of AMACR Expression in Gastric Dysplasia and Adenocarcinoma

Next, we investigated the negative rate of AMACR expression in gastric dysplasia and adenocarcinoma, to evaluate the heterogeneity of AMACR expression. The negative rates of AMACR expression in overall lesions were 15.1 ± 23.9%, 49.0 ± 29.9%, and 59.0 ± 32.2% in low- and high-grade dysplasia and gastric adenocarcinoma groups, respectively (Figure 2). The negative rates of AMACR expression between the three categories were significantly different in the Kruskal-Wallis test (*p* < 0.001). The negative rate of low-grade dysplasia was significantly lower than that of high-grade dysplasia and gastric adenocarcinoma (*p* < 0.001 and *p* < 0.001, respectively). However, there was no significant difference in the negative rate between high-grade dysplasia and adenocarcinoma (*p* = 0.256). 

### 3.3. AMACR Expression in Normal Mucosa Adjacent to the Lesion

AMACR expression was evaluated in the normal mucosa adjacent to the lesion. Detailed analyses were conducted based on the distance from the lesion, categorizing distances as within 2 mm and beyond 2 mm. AMACR expression was observed in 19 of 79 cases within 2 mm (24.1%) and in 14 of 79 cases more than 2 mm away from the lesion (17.7%) (Table 4). However, no significant correlation was found between AMACR expression in gastric lesions and the adjacent normal mucosa. In addition, AMACR expression in the normal mucosa showed no significant difference with the distance from the lesions.

## 4. Discussion

AMACR has been widely used since it was first reported as a novel molecular marker for prostate carcinoma [4]. The diagnostic and prognostic applications of AMACR in various cancers, including prostate cancer, have been explored. However, there is no better utilization than in prostate carcinoma. In daily practice, the differential diagnosis of various gastric lesions, including reactive changes, dysplasia, and adenocarcinoma, is often conducted through the analysis of small biopsy specimens. However, distinguishing among these conditions in such specimens can be challenging. In addition, there are no biomarkers, such as CEA or CA19-9, that can help differentiate adenocarcinoma [12]. Ancillary tests, including immunohistochemistry, can be helpful in the differential diagnosis. Although immunohistochemistry, including p53 and Ki-67, has been performed, distinguishing between gastric lesions can sometimes remain ambiguous. Attempts have been made to develop diagnostic markers, including AMACR, for application in daily practice [3,7,8]. The present study aimed to evaluate the potential diagnostic role of AMACR immunohistochemistry in gastric lesions. In addition, an analysis of the heterogeneity of AMACR expression within gastric lesions was attempted. The results of our study are as follows: (1) Distinct AMACR expression patterns were observed across different types of gastric lesions. (2) Low-grade dysplasia predominantly exhibited a luminal expression pattern of AMACR. (3) AMACR expression demonstrated more significant heterogeneity in high-grade dysplasia and adenocarcinoma than in low-grade dysplasia. (4) The rate of AMACR expression loss was significantly lower in low-grade dysplasia than in high-grade dysplasia and adenocarcinoma.

The present study evaluated AMACR expression in a hotspot (area = 0.785 mm^2^) in the normal mucosa adjacent to the lesion. Given the low positivity rate in normal mucosa, an evaluation of a hotspot was conducted to assess the heterogeneity or extent of expression. This evaluation was based on proximity to the lesion, which was categorized as within 2 mm and beyond 2 mm. The AMACR-positive rates were 24.1% within 2 mm of the lesions and 17.7% for areas beyond 2 mm. These findings contrast with those of previous reports [3,7], which indicated that AMACR was not expressed in normal gastric mucosa. However, Lee found a 4.5% expression rate of AMACR in the non-neoplastic epithelium [8], and Cho et al. observed a focal expression of AMACR in the gastric mucosa with intestinal metaplasia at 7.7% [7]. This discrepancy may have been caused by the interpretation of AMACR expression. Nonetheless, our results provide valuable insights into the understanding of AMACR expression in normal gastric mucosa. Cho et al. defined AMACR expression as focal or diffuse, with focal positivity criteria between 5% and 50% [7]. However, if there is heterogeneity in AMACR expression, it may be more difficult to determine it in small samples according to Cho’s criteria [7]. The significance of AMACR expression in small areas is particularly relevant in biopsy specimen analyses. By adopting a hotspot evaluation method, our findings offer practical insights for analyzing small biopsied tissues. We also examined AMACR expression across the entire normal tissue surrounding the lesion and found no cases of positivity in the whole sections. However, our approach differs from that of previous studies because we focused on hotspot evaluation. Conversely, in the colonic normal mucosa, AMACR expression was identified in 74.5% of cases [13], although these studies did not specify positive criteria based on distribution. Our results suggest that AMACR expression in biopsy samples should be interpreted with caution.

Cho et al. did not categorize adenomas based on low- or high-grade dysplasia [7]. Furthermore, in Lee’s study, differentiation of results by grade was absent [8]. In contrast with our findings, Huang et al. reported that low-grade dysplasia showed weakly positive AMACR expression in only 1 out of 20 cases [3]. The most significant difference from previous results is that AMACR positivity was 80% in low-grade dysplasia. Huang et al. observed positive AMACR expression for high-grade dysplasia in 16 out of 24 cases (64%), aligning closely with our results [3]. Previous studies predominantly focused on the cytoplasmic pattern of AMACR expression. In contrast, our evaluation included both the luminal and cytoplasmic expression patterns, as described above. Notably, we found that the luminal expression pattern is predominant in low-grade dysplasia. Although Cho et al. described the luminal expression pattern, they did not provide specific results [7]. Our findings, highlighting the prevalence of luminal expression in low-grade dysplasia, offer a novel perspective that could facilitate differentiation between low- and high-grade dysplasia.

To determine the heterogeneity of AMACR expression in each lesion, we analyzed both the positivity of hotspots and the negativity rate of lesions. In previous studies, AMACR positivity has been quantified based on the percentage of all positive lesions [3,7,8]. In these reports, it was not clear what the negative rate of AMACR expression was, and it was not possible to infer from these results the heterogeneity of AMACR expression. This can be a major challenge when interpreting AMACR positivity in biopsy samples. This approach allowed us to compare hotspot evaluations with overall expression patterns, providing a more nuanced understanding of the AMACR distribution. For low-grade dysplasia, high-grade dysplasia, and adenocarcinoma, there were four (20.0%), two (10.5%), and nine (22.5%) cases, respectively, which were negative in the whole lesion but showed positivity in hotspots. Through hotspot analysis, the positivity rates were found to be 100.0% in low-grade dysplasia, 62.3% in high-grade dysplasia, and 65.0% in adenocarcinoma. These findings suggest that the diagnostic implication of AMACR immunohistochemistry may be limited to biopsy samples when considering all lesions. Furthermore, we examined the negative rate of AMACR expression by calculating and comparing averages across different lesion types. In high-grade dysplasia and adenocarcinoma, the negative rates were notably higher (49.0 ± 29.9% and 59.0 ± 32.2%, respectively) than in low-grade dysplasia (15.1%). This disparity suggests that AMACR expression loss is less prevalent in low-grade dysplasia, making it a rare finding in biopsy specimens from such lesions. Conversely, approximately half of high-grade dysplasia or adenocarcinoma cases may not exhibit AMACR expression, highlighting the potential variability in the expression patterns between biopsied and resected specimens. Moreover, distinguishing expression patterns, such as luminal versus cytoplasmic patterns, is crucial in the diagnostic process of biopsied specimens.

The negative rate of AMACR expression in colonic adenomas is approximately 25% [14]. Another study reported positive results in 63.7% of cases [13]. Brahim et al. analyzed AMACR expression in colon adenomas and categorized cases into low- and high-grade dysplasia [15]. In their study, five out of six cases of low-grade dysplasia exhibited positive AMACR expression [15]. However, their analysis included only one high-grade dysplasia case [15]. Brahim’s report included only a small number of cases because they were related to inflammatory bowel disease. Furthermore, AMACR expression has also been studied in dysplasia associated with Barrett’s esophagus [9,10]. However, it is not sufficient to understand the diagnostic value of AMACR expression in gastric dysplasia. 

Several studies have highlighted the diagnostic significance of the AMACR in the stomach [3,7,8]. Moreover, the clinicopathological significance and prognostic relevance of AMACR in gastric cancer have also been documented. In Lee’s study, AMACR expression was associated with tumor depth and TNM stage, with a higher positive rate in early gastric cancer than in advanced stages [8]. Because this study included a small number of patients, a large-scale study will be needed for detailed assessment.

Notably, AMACR expression is the lowest in Stage IV gastric cancers [8]. Morz et al. demonstrated that AMACR expression in gastric adenocarcinoma is correlated with poor disease-free survival [16]. Lin et al. identified a similar correlation between AMACR expression and poor prognosis in colorectal cancer [13]. Conversely, Shi et al. reported no significant correlation between AMACR expression and prognosis in colorectal cancer [17]. These divergent findings underscore the need for further research to elucidate the prognostic relevance of AMACR in the gastrointestinal tract, including in the stomach.

## 5. Conclusions

In conclusion, AMACR is a useful diagnostic marker for differentiating low-grade dysplasia from high-grade dysplasia and gastric adenocarcinoma. However, the positivity of AMACR has no diagnostic value in the differentiation of various gastric lesions. Luminal and cytoplasmic expression patterns, and the extent of expression loss, play crucial roles in this differentiation process.

## Figures and Tables

**Figure 1 medicina-60-01475-f001:**
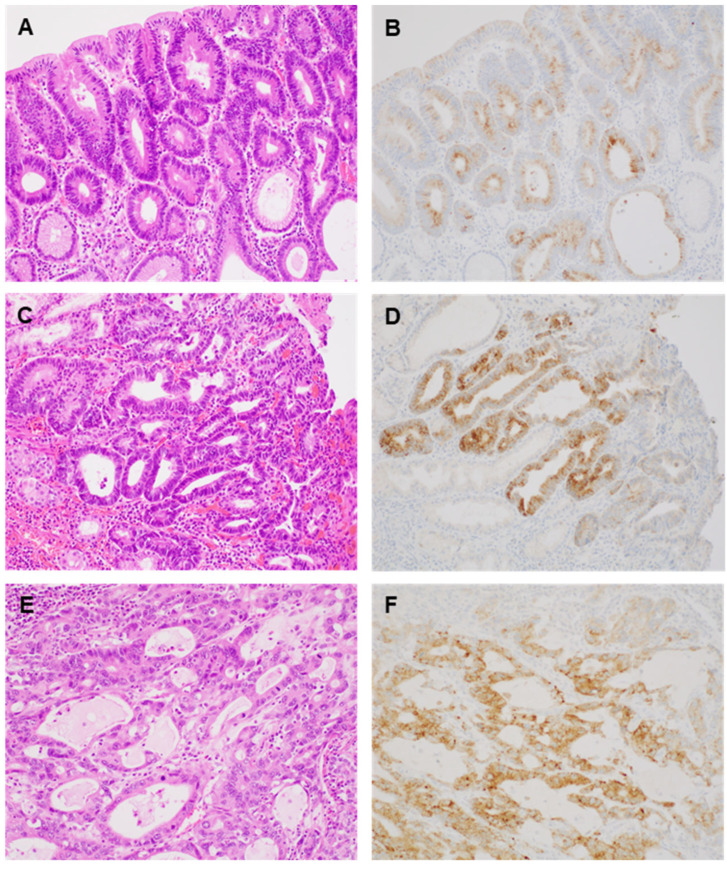
Representative images showing hematoxylin and eosin and AMACR immunohistochemical staining ((**A**,**B**): low-grade dysplasia; (**C**,**D**): high-grade dysplasia; (**E**,**F**): adenocarcinoma) (×200).

**Figure 2 medicina-60-01475-f002:**
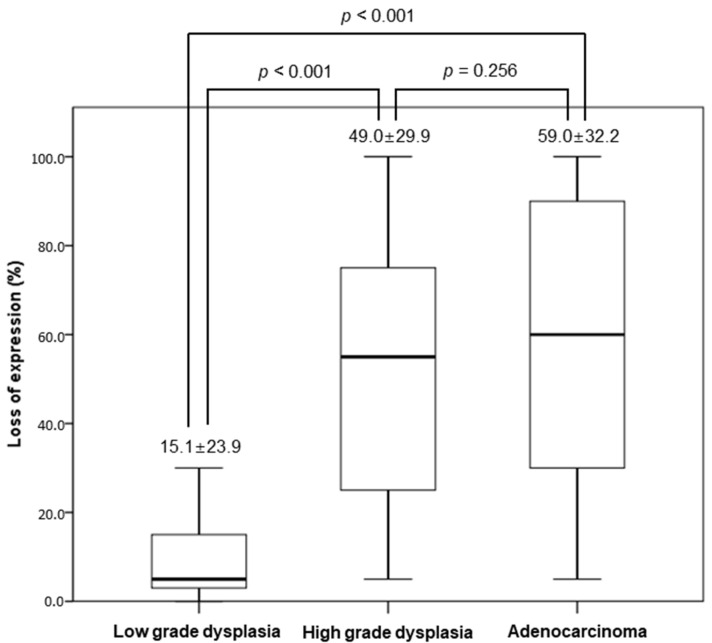
Negative rates of AMACR expression in low- and high-grade dysplasia and adenocarcinoma.

**Table 1 medicina-60-01475-t001:** Patterns of AMACR expression in various gastric lesions.

	AMACR Negative	AMACR Positive		*p*-Value
		Luminal and/or Cytoplasmic	Cytoplasmic Only	
Total (*n* = 79)	36 (45.6)	18 (22.8)	25 (31.6)	
Low-grade dysplasia (*n* = 20)	4 (20.0)	14 (70.0)	2 (10.0)	<0.001
High-grade dysplasia (*n* = 19)	9 (47.4)	3 (15.8)	7 (36.8)	
Adenocarcinoma (*n* = 40)	23 (57.5)	1 (2.5)	16 (40.0)	

Numbers in parentheses represent percentages.

**Table 2 medicina-60-01475-t002:** Correlations between AMACR expression and clinicopathological parameters.

	AMACR Expression		*p*-Value
	Positive	Negative	
Low-grade dysplasia (*n* = 20)	16 (80.0)	4 (20.0)	
Age (years)	64.69 ± 9.08	75.25 ± 5.56	0.041
Sex Male Female	11 (68.8) 5 (31.3)	4 (100.0) 0 (0.0)	0.530
Tumor size (mm)	16.56 ± 6.98	25.75 ± 5.12	0.025
High-grade dysplasia (*n* = 19)	10 (52.6)	9 (47.4)	
Age (years)	67.30 ± 8.04	69.00 ± 10.14	0.689
Sex Male Female	6 (60.0) 4 (40.0)	6 (66.7) 3 (33.3)	1.000
Tumor size (mm)	12.60 ± 7.75	8.89 ± 6.51	0.277
Adenocarcinoma (*n* = 40)	17 (42.5)	23 (57.5)	
Age (years)	69.00 ± 7.71	69.65 ± 8.79	0.808
Sex Male Female	13 (69.8) 4 (30.2)	20 (83.3) 3 (16.7)	0.432
Tumor size (mm)	28.59 ± 16.52	23.39 ± 11.89	0.254

Numbers in parentheses represent percentages.

**Table 3 medicina-60-01475-t003:** Comparison of AMACR expression between hotspot and overall area.

	AMACR Positivity		Discrepancy
	Hotspot *	Overall	
Low-grade dysplasia (*n* = 20)	20 (100.0)	16 (80.0)	4 (20.0)
High-grade dysplasia (*n* = 19)	12 (62.3)	10 (52.6)	2 (10.5)
Adenocarcinoma (*n* = 40)	26 (65.0)	17 (42.5)	9 (22.5)

Numbers in parentheses represent the percentages. * area = 0.785 mm^2^.

**Table 4 medicina-60-01475-t004:** AMACR expression at a hotspot in normal mucosa adjacent to the lesion.

	AMACR Expression in Lesion *		*p*-Value
	Positive	Negative	
Normal mucosa adjacent to the lesion (*n* = 79)			
within 2 mm Positive Negative	10 (23.3) 33 (76.7)	9 (25.0) 27 (75.0)	1.000
away 2 mm Positive Negative	6 (14.0) 37 (86.0)	8 (22.2) 28 (77.8)	0.386

Numbers in parentheses represent percentages. * lesions, including low- and high-grade dysplasia and adenocarcinoma.

## Data Availability

The datasets generated and/or analyzed during the current study are available from the corresponding author upon reasonable request.

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
