# Peer review of "Diagnostic Roles of α-Methylacyl-CoA Racemase (AMACR) Immunohistochemistry in Gastric Dysplasia and Adenocarcinoma"

_medicina, 2024, doi:10.3390/medicina60091475_

Round 1
Reviewer 1 Report
Comments and Suggestions for Authors
-It is recommended that the authors consider revising the title to enhance its conciseness and impact, such as "Diagnostic Utility of α-Methylacyl-CoA racemase (AMACR) immunohistochemistry in gastric neoplasms."
(Additional comments are available in PDF file.)

-
Author Response
Reviewer 1.
-It is recommended that the authors consider revising the title to enhance its conciseness and impact, such as "Diagnostic Utility of α-Methylacyl-CoA racemase (AMACR) immunohistochemistry in gastric neoplasms."
Commented [D1]: It is recommended the authors specify that which type of conjugate is attached to the primary antibody used for immunostaining.
Response) We were unable to find specific information in the manual.
Commented [D2]: The authors should explain about the methods used for measuring the intensity of protein expression, percentage of positively stained cells and rates of loss of AMACR expression.
Response) Two pathologists evaluated the positivity of AMACR expression by the naked eye under the microscope. We added the comment in the revised manuscript.
Commented [D3]: The authors should exactly define the hotspot in this study. What is mean that areas with 0.785 mm2 are considered hotspots?
Response) Hotspot area was selected as the highest level of immunoreactivity through H&E slide after scanning at medium power (100×). We added the comment in the revised manuscript.
Commented [D4]: It seems some mentioned parameters including intensity of protein expression and immunoreactive score (IRS) are not employed in the result section.
Response) Intensity and IRS were used to evaluate AMACR positivity and negativity and were not used for detailed analysis.
Commented [D5]: The results show that there is significant correlation regarding age and tumor size of AMACR expression in AMACR-positive and AMACR-negative patients.
Response) We corrected the sentence from the results of Table 2.
Commented [D6]: The authors said that “In low-grade dysplasia cases, the AMACR expression rate reached 100%” Is this expression rate related to the hotspots? If yes, please clarify in the text.
Moreover, it seems that it is better to use negative AMACR expression rate instead of AMACR expression rate for low-grade dysplasia cases.
Response) As a recommendation, we corrected the sentence.
Commented [D7]: Is “loss rates of AMACR expression” is the same with negative rates of AMACR expression? The authors should use identical terminology in through the text.
Response) As a recommendation, we changed the terminology in through the text.
Commented [D8]: Table 4 is confusing. It is recommended that the table is revised.
Response) As a recommendation, we added the comments to improve the understanding.
Commented [D9]: Are there hotspots in normal mucosa?
Response) The results were obtained at hotspots in normal mucosa..
Commented [D10]: It is recommended that Discussion is rewritten more clearly with more details.
Response) We corrected the discussion part to clarify.
Commented [D11]: AMACR has been widely used for what?
Response) As a description, AMACR is used as the diagnostic and prognostic marker in various cancers in daily practice.
Commented [D12]: The sentence “Beyond its application in prostate cancer” is not clear. The authors should determine the types of application in this sentence.
Response) As a recommendation, we corrected the sentence.
Reviewer 2 Report
Comments and Suggestions for Authors
A very interesting research paper, however:
1. In Table 1, we think that you should use Fisher's exact test since the frequency in 2 cells of the table is less than 5.
2. This Fisher's exact test needs a post-hoc analysis, your test of independence can evaluate the statistic difference overall, but not between each group pairwise-please take this into account.
3. Table 1 misses the percentages in brackets.
4. Why the scoring system was developed just to establish if the sample was negative or positive? in your scoring system a sample that has 4 points meaning if he has moderate staining in 50 percent cases he is negative to AMACR expression? this is a bit confusing.
5. A Kruskal-Wallis test should be performed to see if the scores are significantly different between the three categories.
6. It would be interesting to see if IRS correlates with tumor markers like CEA or CA 19-9. For more information see this: https://doi.org/10.3390/ijms25147898
7. Lines 139-140 you said that the clinical-pathological features did not correlate with AMACR expression, however table 2 in the low-grade dysplasia results show that older patients with larger tumors have a higher incidence of negative AMACR.
8. Patient number is still rather small and lack healthy (or non-dysplasia/adenocarcinoma) controls.
Comments on the Quality of English LanguageMinor English language revision
Author Response
Reviewer 2.
A very interesting research paper, however:
- In Table 1, we think that you should use Fisher's exact test since the frequency in 2 cells of the table is less than 5.
Response) As a recommendation, the Fisher’s exact test was performed.
- This Fisher's exact test needs a post-hoc analysis, your test of independence can evaluate the statistic difference overall, but not between each group pairwise-please take this into account.
Response) As a recommendation, the Fisher’s exact test was performed between two groups. There were significant differences in AMACR expression pattern between low-grade dysplasia and high-grade dysplasia and between low-grade dysplasia and adenocarcinoma (P = 0.003 and P < 0.001, respectively). However, there was no significant difference between high-grade dysplasia and adenocarcinoma (P = 0.197). We added these results in the revised manuscript.
- Table 1 misses the percentages in brackets.
Response) We corrected the Table 1.
- Why the scoring system was developed just to establish if the sample was negative or positive? in your scoring system a sample that has 4 points meaning if he has moderate staining in 50 percent cases he is negative to AMACR expression? this is a bit confusing.
Response) The immunostaining intensity must be at least moderate to be considered positive. Also, the assessment is based on the requirement that at least a quarter of the range should be positive. These criteria may vary by researchers. These criteria will need to be validated with many cases in the future.
- A Kruskal-Wallis test should be performed to see if the scores are significantly different between the three categories.
Response) As a recommendation, the Kruskal-Wallis test was performed. The loss rates of AMACR expression between three categories were significantly different in the Kruskal-Wallis test (P < 0.001). This result is added in the revised manuscript.
- It would be interesting to see if IRS correlates with tumor markers like CEA or CA 19-9. For more information see this: https://doi.org/10.3390/ijms25147898
Response) We added the information of recommended article in the revised manuscript.
- Lines 139-140 you said that the clinical-pathological features did not correlate with AMACR expression, however table 2 in the low-grade dysplasia results show that older patients with larger tumors have a higher incidence of negative AMACR.
Response) We corrected the sentence from the results of Table 2.
- Patient number is still rather small and lack healthy (or non-dysplasia/adenocarcinoma) controls.
Response) As pointed out, our cases are small in number. Our study can be considered preliminary, and a large-scale study could be conducted based on our results. As more cases are accumulated, we plan to conduct a large-scale study. This limitation is described in the revised manuscript.
Round 2
Reviewer 2 Report
Comments and Suggestions for Authors
We congratulate the authors for the extraordinary work!